# Exposure to Radiofrequency Induces Synaptic Dysfunction in Cortical Neurons Causing Learning and Memory Alteration in Early Postnatal Mice

**DOI:** 10.3390/ijms25168589

**Published:** 2024-08-06

**Authors:** Ju Hwan Kim, Jun Young Seok, Yun-Hee Kim, Hee Jung Kim, Jin-Koo Lee, Hak Rim Kim

**Affiliations:** 1Department of Pharmacology, College of Medicine, Dankook University, Cheonan 31116, Republic of Korea; jhkim731@dankook.ac.kr (J.H.K.); seogant95@naver.com (J.Y.S.); leejking@dankook.ac.kr (J.-K.L.); 2Department of Biology Education, Institute of Agriculture and Life Science (IALS), Gyeongsang National University, Jinju 52609, Republic of Korea; cefle@gnu.ac.kr; 3Department of Physiology, College of Medicine, Dankook University, Cheonan 31116, Republic of Korea; heejungkim@dankook.ac.kr

**Keywords:** radiofrequency electromagnetic fields, cerebral cortex, synapse, cell adhesion molecules, cyclin-dependent kinase 5, spatial learning and memory

## Abstract

The widespread use of wireless communication devices has necessitated unavoidable exposure to radiofrequency electromagnetic fields (RF-EMF). In particular, increasing RF-EMF exposure among children is primarily driven by mobile phone use. Therefore, this study investigated the effects of 1850 MHz RF-EMF exposure at a specific absorption rate of 4.0 W/kg on cortical neurons in mice at postnatal day 28. The results indicated a significant reduction in the number of mushroom-shaped dendritic spines in the prefrontal cortex after daily exposure for 4 weeks. Additionally, prolonged RF-EMF exposure over 9 days led to a gradual decrease in postsynaptic density 95 puncta and inhibited neurite outgrowth in developing cortical neurons. Moreover, the expression levels of genes associated with synapse formation, such as synaptic cell adhesion molecules and cyclin-dependent kinase 5, were reduced in the cerebral cortexes of RF-EMF-exposed mice. Behavioral assessments using the Morris water maze revealed altered spatial learning and memory after the 4-week exposure period. These findings underscore the potential of RF-EMF exposure during childhood to disrupt synaptic function in the cerebral cortex, thereby affecting the developmental stages of the nervous system and potentially influencing later cognitive function.

## 1. Introduction

The widespread use of wireless communication devices in recent decades has resulted in increased exposure to radiofrequency electromagnetic fields (RF-EMF). As mobile phone use typically requires close contact with the head, its possible effects on the central nervous system are of particular concern [1]. Despite many controversies, there is accumulating evidence regarding the biological effects of RF-EMF exposure on the central nervous system (CNS), such as changes in intracellular calcium homeostasis, neuronal damage, and neurotransmitter disruptions [2,3,4].

In particular, the exposure of children to RF-EMF has increased rapidly in recent years. Mobile phone use among children has become a major determinant of RF-EMF exposure [5,6]. Previous studies demonstrated that the specific absorption rate (SAR) of 5-year-old children is twice that of 20-year-old adults [7]. Therefore, RF-EMF exposure may have a greater impact on children during the developmental stages of the CNS.

We previously reported the effects of RF-EMF exposure on hippocampal neurons in early postnatal mice. This study revealed that RF-EMF exposure leads to changes in the structure of postsynaptic connections and impedes neurite outgrowth in hippocampal neurons, specifically leading to a decline in recognition memory [8]. Research has highlighted the potential concerns regarding the impact of RF-EMF on neural development in young mice [8].

The cerebral cortex is a highly developed region situated on the surface of the brain [9]. The neocortex, a crucial part of the cerebral cortex in mammals, is involved in a range of essential functions including sensory perception, motor control, higher cognitive processes, and complex behaviors. It plays a pivotal role in integrating sensory information, facilitating executive functions such as decision-making and problem-solving and supporting language and communication skills [10]. Dysfunction of the cerebral cortex may be associated with neurodegenerative diseases such as Alzheimer’s disease [11]. Additionally, young children with impaired cortical development may have neurodevelopmental disorders such as attention deficit hyperactivity disorder and autism spectrum disorders (ASDs) [12,13]. However, it remains unclear how RF-EMF exposure affects the cerebral cortex during neurodevelopment, despite significant societal concerns about this issue. Although cortical dysfunction and alterations in synaptic connections are important aspects of brain disorders, a comprehensive understanding requires considering the interactions between neurotransmitter systems, neurodevelopmental processes, brain connectivity, inflammation, genetic and epigenetic factors, and environmental influences. Each of these components contributes to the complex etiology and manifestation of various brain-related diseases, including Alzheimer’s disease, autism spectrum disorder, attention deficit hyperactivity disorder, etc.

In this study, we investigated the influence of RF-EMF exposure on synapse development in cerebral cortical neurons of mice. Specifically, we examined the changes in pivotal genes and proteins involved in synaptic formation, including neurexin, neuroligin, and cyclin-dependent kinase 5 (CDK5). Furthermore, we investigated the effect of RF-EMF exposure on neurological behaviors associated with spatial learning and working memory in early postnatal mice. The exposure regimen involved daily exposure to 1850 MHz RF-EMF at a SAR of 4.0 W/kg for 5 h per day over a period of 4 weeks.

## 2. Results

### 2.1. Alterations in the Number of Dendritic Spines in Cortical Neurons of Mice by RF-EMF Exposure

To investigate the effects of RF-EMF exposure on neural development and synapse formation, the dendritic spines (DSs) of neurites in the cerebral cortex of RF-EMF-exposed mice were analyzed.

After exposure to 1850 MHz RF-EMF for 5 h a day at a SAR of 4.0 W/kg for 4 weeks, the mouse brain was isolated. The effects of RF-EMF exposure on neurodevelopment and synapse formation were investigated using transmission electron microscopy (TEM) to study cerebral cortex sections from isolated brains. DSs can be classified into three types: mushroom, thin, and stubby [8]. Representative TEM images of dendritic spines in the cerebral cortex of control and RF-EMF-exposed mice are shown in Figure 1A.

The total number of DSs significantly decreased in the cerebral cortex (Figure 1B(a)). Specifically, the number of mushroom-type and thin-type DSs showed statistically significant decreases in the cerebral cortexes of mice after RF-EMF exposure (Figure 1B(b,d)); however, the number of stubby DSs remained unchanged (Figure 1B(c)). Collectively, these findings indicate a significant decrease in the overall number of DSs within the cortex of postnatal mice after RF-EMF exposure.

### 2.2. Effect of RF-EMF Exposure on Synaptic Maturation in Cortical Neurons of Mice

Postsynaptic density protein-95 (PSD95) is a scaffolding protein that regulates the synaptic localization of many receptors, channels, and signaling proteins and is a major regulator involved in the maturation of excitatory synapses by interacting and trafficking N-methyl-D-aspartic acid receptors and α-amino-3-hydroxy-5-methyl-4-isox-azoleproprionic acid receptors to the postsynaptic membrane [14]. Therefore, PSD95 is a commonly used marker of postsynaptic sites in the nervous system.

We cultured cortical neurons derived from ICR mice on postnatal day 1 (P1) to investigate the effects of RF-EMF exposure on synaptic maturation during synapse development. Primary cultured cortical neurons were exposed to 1760 MHz RF-EMF at a SAR of 4.0 W/kg for 5 h daily over a period of 9 consecutive days. To assess synaptic changes, PSD95 and microtubule-associated protein 2 (MAP2) were stained with specific antibodies (Figure 2A).

The results showed a significant decrease in PSD95 puncta in neurons exposed to RF-EMF compared to the control group at multiple time points: DIV 3 (49.5 ± 10.8, *n* = 6), DIV 5 (65.9 ± 11.7, *n* = 13), and DIV 9 (67.7 ± 3.6, *n* = 7) (Figure 2B,C). Although the difference was not statistically significant at DIV 7 (88.0 ± 8.6, *n* = 14), there was an observable trend suggesting a reduction in PSD95 puncta in RF-EMF-exposed neurons compared to controls. Consequently, subsequent investigations focused on DIV 9. These findings indicated that RF-EMF exposure led to a significant decrease in PSD95 puncta during the early developmental stages of synapses in mouse cortical neurons.

### 2.3. Effect of RF-EMF Exposure on Neurite Outgrowth in Developing Cortical Neurons of Mice

Neurite outgrowth was assessed in terms of total neurite length (μm), the number of branches, and soma size (μm^2^) (Figure 3A). Analysis of neurite length revealed a steady increase from DIV 3 to DIV 9, peaking at DIV 9 (Figure 3B). However, RF-EMF-exposed cultured cortical neurons exhibited a significant reduction in neurite length from DIV 5 to DIV 9 compared to controls. Additionally, the number of branches in RF-EMF-exposed neurons significantly decreased at DIV 7 and DIV 9 relative to that in the controls (Figure 3C). The soma size of RF-EMF-exposed cultured cortical neurons was generally higher than that in controls on DIV 3, 7, and 9, but not on DIV 5 (Figure 3D).

### 2.4. Changes in Synapse Formation-Related Genes and Proteins in the Cerebral Cortex of Mice after RF-EMF Exposure

We further studied whether the morphological changes in synapses due to RF-EMF exposure were accompanied by changes in the expression of genes and proteins involved in synapse formation in the mouse cortex. Synapse formation-related genes such as *neuroligin 2 (nlgn2)*, *neuroligin 3 (nlgn3)*, *and neurexin 1α (nrxn1α)* are pre- and postsynaptic adhesion proteins that maintain synapse formation and connect neurons [15]. Additionally, CDK5 promotes synapse formation by regulating the movement of synaptic vesicles to synapse formation sites [16].

We investigated whether RF-EMF exposure influences the expression *neuroligin 2*, *neuroligin 3*, *neurexin 1α*, and CDK5 in the cerebral cortex. The data showed significant decreases in the expression levels of *neuroligin 2*, *neuroligin 3*, *and neurexin 1α* genes in mice exposed to RF-EMF (Figure 4A(a–d)). Additionally, the protein expression of CDK5 was reduced in the cerebral cortex following RF-EMF exposure (Figure 4B(a,b)).

### 2.5. Spatial Learning and Memory Function in Young Mice Affected by RF-EMF Exposure

To elucidate whether changes in synapse formation and function in the cerebral cortexes of RF-EMF-exposed mice affected their learning and memory functions, the spatial learning and memory functions of RF-EMF-exposed mice were evaluated using the Morris water maze test. The mice completed training before behavioral testing. After training, five actual trials were conducted. Swimming distance (cm) and escape latency (s) were measured using a tracking program.

This study showed that there were no significant differences in the total distance or escape latency between control mice and RF-EMF-exposed mice until the fourth trial (Figure 5A,C). Furthermore, while the total distance traveled and escape latency gradually decreased over five trials in sham control mice, no significant change was observed in RF-EMF-exposed mice (Figure 5B,D). Notably, mice exposed to RF-EMF exhibited total-distance-traveled and escape-latency values that were more than twice as high as those of control mice (Figure 5). These findings suggest that RF-EMF exposure (1850 MHz, 4 W/kg, 5 h daily for 4 weeks) may affect spatial learning and memory in early postnatal mice.

## 3. Discussion

This study investigated the potential effects of RF-EMF exposure on synapse formation and function in the cortical neurons of the mouse cerebral cortex, along with its impact on associated genes and proteins. Additionally, we studied how these changes influenced spatial learning and memory in mice.

Although several studies are in progress, and a clear mechanism has not yet been revealed yet, several putative mechanisms have been proposed. RF-EMF can cause localized heating of tissues, potentially affecting neural tissue if exposure is high enough [17,18]. Also, some studies suggest that RF-EMF might influence CNS function through oxidative stress [19,20], neurotransmitter release alteration [21,22], gene expression [23], inducing weak electrical currents in neural tissues [24], affecting neuronal cell proliferation or apoptosis [25], and impacting blood–brain barrier permeability [26]. While there are numerous hypotheses about how RF-EM fields might influence CNS development and function [27], definitive conclusions are challenging due to variability in study designs, exposure levels, and biological responses. More research is needed to clarify these mechanisms and their implications for health.

In this study, the frequency of RF-EMF was determined according those to commonly used in actual telecommunication services. Fourth-generation (4G) communication systems operate globally within the frequency range of 600 MHz to 2.5 GHz for Long-Term Evolution (LTE) services. In this study, a frequency of 1850 MHz was studied, as it corresponds to one of the most widely used bands for 4G LTE communication services in the Republic of Korea. In addition, the 4 W/kg SAR value that we used in this study is the maximum permitted SAR exposure to normal users, standardized based on SAR-related international organizations and major countries (the SAR standards of SAR-related international organizations and major countries, such as the National Radio Research Agency; https://rra.go.kr/en/sar/standard.do (accessed on 30 July 2024).

Initially, we observed a significant reduction in the number of DSs, particularly mushroom-type DSs, which are known for their strong synaptic signaling, in the cerebral cortexes of early postnatal mice following RF-EMF exposure (Figure 1). This finding is consistent with that of our previous study on the hippocampus, indicating that RF-EMF exposure decreases the DSs in various brain regions [8]. Such alterations in mushroom-type DSs suggest the inhibition of synaptic formation and changes in synaptic function in the cortical neurons of young mice.

Subsequently, we assessed the effect of RF-EMF exposure on synaptic density during synapse development by examining PSD95, a key regulator of synaptic plasticity and dendritic markers [28]. Our results revealed a significant decrease in PSD95 puncta in the cortical neurons of RF-EMF-exposed mice (Figure 2), indicating the reduced formation of dendrites in cortical neurons.

Furthermore, RF-EMF exposure led to diminished neurite outgrowth in cultured cortical neurons, as evidenced by the reduced neurite length and branching assessed via MAP2 staining (Figure 3). These findings suggest that RF-EMF exposure alters the synaptic structure and inhibits synaptic density and neurite outgrowth in cortical neurons.

Given the observed inhibitory effect of RF-EMF exposure on synapse formation, we investigated its impact on synaptic cell adhesion molecules, including *neuroligin 2*, *neuroligin 3*, *and neurexin 1α*, which are critical for synapse maintenance and neuronal connectivity. Our data indicate a decrease in the expression of these genes in the prefrontal cortexes of RF-EMF-exposed mice (Figure 4A).

During brain development, the precise connection between neurons involves synaptic cell adhesion molecules, such as neurexins and neuroligins, which are pivotal for synapse development, function, and plasticity [29,30]. Dysfunction of these genes has been associated with various cognitive disorders, including ASDs, schizophrenia, and intellectual disabilities [30,31,32,33,34,35,36,37,38], underscoring their role in the pathogenesis of these conditions.

Moreover, we observed a decrease in CDK5 protein levels in mouse cortexes following RF-EMF exposure (Figure 4B). CDK5 is crucial for various aspects of neuronal development, including synaptic growth, maturation, dendritic spine formation, and synaptic plasticity [39,40]. Notably, CDK5 has been implicated in the regulation of learning and memory processes [41], as evidenced by severe spatial learning impairments in CDK5-deficient mice [42].

Behavioral assessments using the Morris water maze corroborated these findings, demonstrating impaired spatial learning and memory in young RF-EMF-exposed mice (Figure 5). This task has been widely utilized to study the psychological processes and neural mechanisms underlying spatial learning and memory in rodents [43]. In a previous study, we measured object memory using a novel object recognition (NOR) behavioral task and showed that the memory index was significantly reduced in RF-EMF-exposed young mice [8]. The NOR task is used to evaluate cognition, particularly recognition memory in rodents, and it does not allow for measures of learning; however, the Morris water maze can be used to measure spatial learning.

Our study focused on the prefrontal cortex of the cerebral cortex, a region crucial for executive function and cognitive processes [44,45,46], including rule learning [47]. Dysfunction in the development and maturation of this region has been implicated in conditions such ASD, characterized by deficits in socialization, communication, and repetitive behaviors [13].

## 4. Materials and Methods

### 4.1. Animals

ICR pups (postnatal day (P) 0) and dams were purchased from Samtako BioKorea (Osan, Republic of Korea). Mice were maintained under specifically controlled conditions (am-bient temperature, 23 ± 2 °C; 12-h light/dark cycle). Pups were fed breast milk from their mothers, which were supplied with food pellets and water ad libitum. All procedures complied with the National Institutes of Health guidelines for animal research and were approved by the Dankook University Institutional Animal Care and Use Committee (IACUC; DKU-15-001, 14 April 2015), which adheres to the guidelines issued by the Institution of Laboratory of Animal Resources.

The number of postnatal ICR pups were usually about 10–12, and they were exposed to RF-EMF with their mothers for 3 weeks. After 3 weeks of feeding, the dams were separated from their pups, and only the pups continued to be exposed to RF-EMF for an additional week. To prevent killing the pups by their mother, we avoided touching the pups directly and did not stress the dams by providing enough food pellets and water. The number of pups was randomly matched, and the same number of pups were provided to each dam to minimize the weight difference of the pup between each group. Litters were not gender-balanced. Furthermore, since pups were in young infancy, it was considered that there would be no difference experimentally, and all experiments were conducted without reference to gender.

### 4.2. RF-EMF Exposure in Mice

Mice were exposed to RF-EMF using the Wave Exposer V20 described in a previous study [8]. All the experiments were conducted at our animal facility, which was maintained at a constant temperature. It was confirmed that the RF-EMF generator created a 1850 MHz signal using a measuring spectrum analyzer (NS-30A) (LIG Nex, Gyeonggi-do, Republic of Korea). Subsequently, the SAR value was estimated to be 4.0 W/kg by a 0.0001 °C resolution temperature sensor by measuring temperature changes in the saline water of the mouse phantom exposed to a 1850 MHz continuous wave (CW) without modulation. The temperature change in saline water was measured by a 0.0001 °C resolution in this research to obtain a more precise SAR value with a finer temperature measurement system (FLUKE 1586A). The SAR value in the central position of the mouse phantom was also acquired by numerical analysis by Ansys HFSS 13. Our measurement of the RF signal and SAR value generated from our RF-EMF generator produced 1850 MHz RF-EMF with 4.0 W/kg SAR.

Pups and dams received whole-body exposure to 1850 MHz RF-EMF at a SAR value of 4.0 W/kg for 5 h/day for 4 weeks (from P1 to P28). The sham-exposed group was maintained under the same environmental conditions and treated with the same circular pattern as that of the RF-exposed group without RF-EMF exposure. When the RF-EMF exposure group moved to the exposure device located in the animal breeding facility, the control group also moved to the same room in the animal breeding facility and returned to the original breeding room at the same time after exposure to RF-EMF. Sham- and RF-EMF-exposed mice were allowed to move freely in their cages. After the 4-week exposure, mice of either sex were sacrificed for morphological and biochemical studies.

### 4.3. Primary Cultures of Mouse Cortical Neurons

Primary cortical neurons were derived from P1 ICR mouse brains following previously established protocols [8]. The cortex was isolated in a Ca^2+^- and Mg^2+^-free HEPES-buffered Hanks salt solution (HHSS). The isolated tissues were incubated with 0.025% trypsin–HHSS solution for 15 min. Cell dissociation was carried out through trituration using a flame-narrowed Pasteur pipette, and cells were plated on 18 mm round cover glasses. The culture medium comprised Neurobasal medium supplemented with L-glutamine, 2% B27 supplement, 0.25% Glutamax I, and penicillin/streptomycin/amphotericin B. Matrigel (0.2 mg/mL; BD Bioscience, San Jose, CA, USA), which pre-coated the cover glasses. Cortical neurons were cultured under 10% CO_2_ and 90% air at 37 °C, with 75% of the media refreshed on days 3 and 7. Prepared cortical neurons were exposed to 1760 MHz RF-EMF with a SAR value of 4.0 W/kg for 5 h every day for 9 days.

### 4.4. RF-EMF Exposure in Cultured Neurons

Cultured neurons were prepared using an established protocol [8]. During exposure, the incubator temperature was maintained at 37 °C under 5% CO_2_ and 95% air. After exposure to RF-EMF, both cells exposed to RF-EMF and non-exposed cells were simultaneously transferred to an incubator. Then, the cells were immunolabeled at 9 days in vitro (DIV). The generation of this RF-EMF has been described in detail in [8].

### 4.5. Immunocytochemistry

After fixation in cooled methanol at −20 °C, cortical neurons underwent permeabilization with 0.3% Triton X-100 (Sigma-Aldrich, Louis, Mo, USA) for 5 min. Following blocking with 10% BSA, cells were incubated overnight at 4 °C with the following primary antibodies: mouse anti-MAP2 (1:50) (Sigma-Aldrich, St. Louis, MI, USA), rabbit anti-PSD95 (1:100) (Abcam, Cambridge, UK), Alexa Fluor 488-conjugated anti-rabbit IgG, and Alexa Fluor 555-conjugated anti-mouse IgG (ThermoFisher, Rockford, IL, USA); the cells were incubated for 1 h and 30 min at room temperature. The cells were mounted with VECTASHIELD Mounting Medium (Vector Laboratories Inc., Burlingame, CA, USA). Alexa Fluor 488- and Alexa Fluor 555-labeled neurons were visualized using an FV3000 confocal microscope (FV3000, Olympus, Tokyo, Japan).

### 4.6. Confocal Imaging and Morphological Analysis

Immunolabeled cortical neurons were observed as previously described [8]. Total neurite outgrowth length was assessed through optical sections using a 20x objective lens. For protein localization analysis, a series of eight cross-sections were captured at 1 μm intervals along the *z*-axis using a 60× objective lens. These sections were then combined to generate a compressed z-stack. The quantification of PSD95 puncta was performed according to an established protocol described in a previous report [8]. The quantified puncta counts for PSD95 were expressed as the mean ± SEM. Each measurement was based on a specific number of randomly selected image fields per coverslip (n). Morphological change in cultured cortical neurons, including total outgrowth, the number of branches, and soma size, were evaluated using Metamorph software (version 7.7, Molecular Devices, Sunnyvale, CA, USA).

### 4.7. Transmission Electron Microscopy (TEM)

Cerebral cortex obtained from sham- and RF-EMF-exposed mouse brains (*n* = 4) were fixed in Karnovsky fixative (EMS Microscopy Academy, Hatfield, PA, USA) for 4 h at 4 °C, washed three times with 0.1 M phosphate-buffered saline (PBS), and post-fixed with 1% OsO_4_ in 0.1 M PBS for 2 h at 4 °C. After washing with 0.1 M PBS, specimens were dehydrated through a graded 70–100% ethanol series, exchanged with propylene oxide, and embedded in a mixture of Epon 812 and Araldite (Polysciences Inc., Warrington, PA, USA). Ultrathin sections (70 nm) were cut using a Leica Em UC6 Ultramicrotome (Leica, Wetzlar, Germany). A ribbon of ultrathin serial sections from each animal was collected on a Ni grid and stained with uranyl acetate and lead citrate. The sections were collected on TEM nickel grids and observed using a transmission electron microscope (JEM-1400 flash; JEOL, Tokyo, Japan) at 120 kV.

Importantly, each mouse cortex area and layer had various densities. However, to minimize the differences between each experimental group, we tried to secure the same parts as much as possible when preparing electron microscope samples, secured various sections, and randomly acquired multiple electron microscope images. In the images obtained in this way, the dendritic spine was analyzed to ensure that comparison between the RF-EMF exposure group and the control group could occur.

To measure for number of dendritic spines, samples were immediately prepared with control mice (*n* = 4) and RF-EMF exposed mice (*n* = 4). We generated 6–7 images per mouse and counted the number of different types (thin, stubby, and mushroom) of dendritic spines in 28 images of the control group and 25 images of the RF-EMF-exposed group. The average value of each type of dendritic spine measured was shown.

### 4.8. Quantitative RT-PCR

Mice were euthanized by cervical dislocation, and their heads were quickly decapitated with scissors; then, the cerebral cortex was rapidly separated from each brain on ice and the prefrontal cortex area additionally dissected. The qRT-PCR was carried out following the method in [2]. Total RNA was purified from the whole cerebral cortex of each mouse in each group using TRIzol reagent (Thermo Fisher Scientific, Frederick, MD, USA). After RNA quantification, 1 μg of total RNA was used for reverse transcription PCR using AccuPower^®^ CycleScript RT PreMix dT20 (Bioneer, Daejeon, Republic of Korea). Primer sequences were as follows: neuroligin 2—forward 5′-TGTCATGCTCAGCGCAGTAG-3′ and reverse 5′-GGTTTCAAGCCTATGTGCAGAT-3′; neuroligin 3—forward 5′-CCCTGGGCTTCCTCAGTTTG-3′ and reverse 5′-GGCAATGGTACTCTGGCACC-3′; neurexin -1α—forward 5′-ACCGTGCCTTAGCAATCCTTGC-3′ and reverse 5′-GTCGTAGCTCAAAACCGTTGCC-3′. Mouse Gapdh primer was purchased from QIAgen (QIAgen, Hilden, Germany). Quantitative RT-PCR reactions were performed with the Rotor-gene SYBR Green supermix kit (QIAgen, Hilden, Germany), and fluorescence was measured using a Rotor Gene PCR Cycler with Rotor-Gene Q software v. 2.3.1 (QIAgen, Hilden, Germany). Three biologically independent experiments were performed, and each PCR reaction was performed in triplicate. The relative levels of specific mRNAs were calculated by normalization to GAPDH expression by the 2^−ΔΔCt^ method. Semi-quantitative RT-PCR reactions (sqPCR) were carried out using PCR PreMix (Bioneer, Daejeon, Republic of Korea). Subsequently, the sqPCR product of each gene was separated by 1.5% agarose gel electrophoresis, and the signal intensity of each PCR product was visualized following staining of DNA using Syto 60 (Li-Cor, Lincoln, NE, USA) using the Odyssey infrared imaging system (Li-Cor). 

### 4.9. Western Blot

Sham-exposed or RF-EMF-exposed mice were rapidly sacrificed, and prefrontal cortex area was rapidly dissected. The tissue was lysed with RIPA buffer (ThermoFisher, Rockford, IL, USA) supplemented with protease and a phosphate inhibitor cocktail (ThermoFisher, Rockford, IL, USA). Whole lysates were homogenized in ice-cold buffer and briefly sonicated. Protein concentrations were measured using a Bio-Rad DCTM protein assay (Bio-Rad, Hercules, CA, USA). Total protein (20 μg) was separated using 10% sodium dodecyl sulfate-polyacrylamide gel electrophoresis (SDS-PAGE) and transferred with transfer buffer to a polyvinylidene difluoride (PVDF) transfer membrane (Bio-Rad, Hercules, CA, USA). CDK5 and α-tubulin were detected in the membranes using anti-CDK5 antibody (1:1000) (Abcam, Cambridge, UK) and anti-β-actin antibody (1:1000) (Cell Signaling Technology, Danvers, MA, USA), respectively. Protein bands were visualized using the Odyssey infrared imaging system (Li-Cor Biosciences, Lincoln, NE, USA). The intensity of each band was quantified and normalized using α-tubulin as an internal loading control.

### 4.10. Morris Water Maze Test

The Morris water maze test consists of a round tank 120 cm in diameter and 54 cm deep. The water temperature was maintained at 25 °C. A platform (diameter 17 cm) submerged 1.5 cm under the water surface was placed on the center of one of the four imaginary quadrants of the tank and maintained in the same position during all trials. Four different shapes of visual clues were placed on the walls of the water maze room. During the experiments, the tank was videotaped, and the scores for latency required to escape to the platform represented the distance traveled from the starting point to the platform. A recorded video analyzer (Ethovision XT 12 software, Noldus, Wageningen, The Netherlands) was used to measure the distance traveled and the swimming speed of the mouse in the pool. The training session consisted of three consecutive trials during which the mouse was left in the tank facing with wall and allowed to swim freely to the escape platform. If the mouse did not find the platform in 120 s, it was gently guided to it. The test session was performed 24 h later and was similar to the training session, except for the number of trials. In the test session, the mouse was allowed to remain on the platform for 10 s after escaping to it and was then removed from the tank for 20 s before being placed at the next starting point in the tank. This procedure was repeated five times.

According to the literature, ICR mice are known to experience significant visual impairments, which makes them unsuitable for tasks such as the Morris water maze [48]. Therefore, there may be limitations in interpreting the water maze results in our study, and additionally, it may be recommended to employ an alternative cognitive test that does not heavily depend on visual cues to more accurately assess the cognitive deficits induced by RF-EM fields.

### 4.11. Statistical Analysis

Data are presented as the mean ± standard error of the mean (SEM). Student’s *t*-test was used to compare two groups statistically. For comparisons involving more than two groups, a one-way analysis of variance (ANOVA), followed by Bonferroni’s test, was utilized. All statistical analyses were performed using GraphPad Prism 4 (GraphPad Software Inc., San Diego, CA, USA). Statistical significance was determined at *p* < 0.05 and 0.01 levels.

## 5. Conclusions

Overall, our findings suggest that RF-EMF exposure at 4 W/kg SAR for 5 h daily for 4 weeks impairs synapse formation and function, reduces synaptic cell adhesion molecules and CDK5 levels, and ultimately leads to deficits in spatial learning and memory in early postnatal mice. These results underscore the potential developmental implications of RF-EMF exposure, particularly in the context of neurodevelopmental disorders, such as ASD, necessitating further investigation.

## Figures and Tables

**Figure 1 ijms-25-08589-f001:**
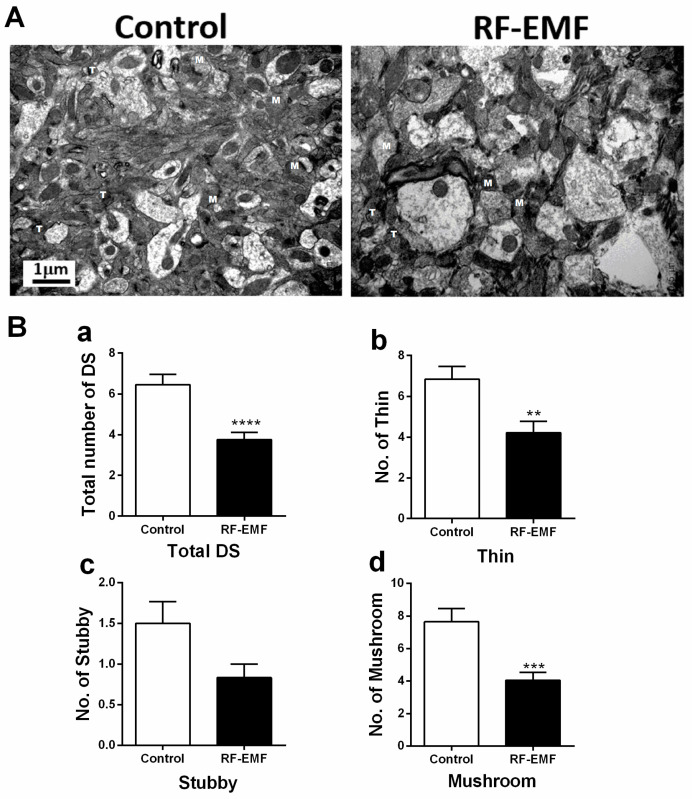
Changes in dendritic spines in the cerebral cortexes of early postnatal mice exposed to RF-EMF. (**A**) Representative transmission electron microscopy images of dendritic spines in control and RF-EMF-exposed mice. (**B**) The number of thin—(**B**(**b**)), stubby—(**B**(**c**)), and mushroom-type (**B**(**d**)) dendritic spines and the total number of dendritic spines (**B**(**a**)) are analyzed. T, thin; M, mushroom. Data are expressed as mean ± SEM. Statistical significance is evaluated using Student’s *t*-test. ** *p* < 0.01, *** *p* < 0.001, and **** *p* < 0.0001 vs. control (control, *n* = 4 mice; RF-EMF, *n* = 4 mice).

**Figure 2 ijms-25-08589-f002:**
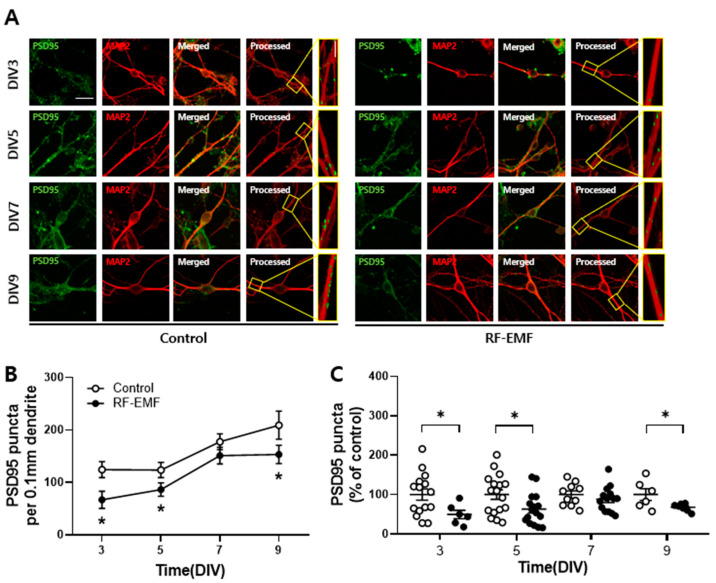
Exposure to RF-EMF decreases the expression of postsynaptic density 95 (PSD95) in cortical neurons of mice. (**A**) Confocal images showing the z-stack of cortical neurons expressing PSD95 (green) and microtubule-associated protein 2 (MAP2) (red). The images of PSD95 and MAP2 are analyzed using Image J software bundled with 64-bit Java 8. (**B**,**C**) A dot graph summarizing the alterations in PSD95 puncta following RF-EMF exposure (white dot, control; black dot, RF-EMF). Data are presented as mean ± SEM. Statistical significance is evaluated by using Student’s *t*-test (**B**,**C**). * *p* < 0.05 vs. control (control, *n* = 6–17 cells; RF-EMF, *n* = 6–16 cells). Scale bar, 20 μm.

**Figure 3 ijms-25-08589-f003:**
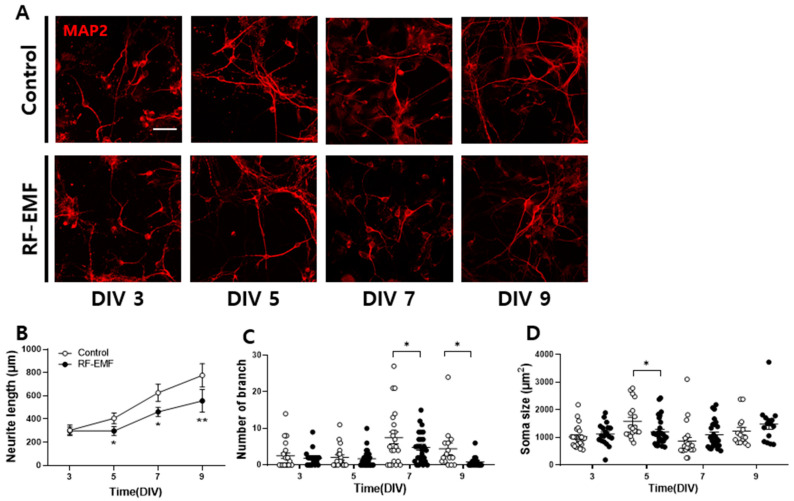
Exposure to RF-EMF decreases neurite outgrowth in developing cortical neurons. (**A**) The neurite outgrowth (MAP2, red) is observed in developing neurons. (**B**) Temporal changes in neurite length. (**C**) Variations in the number of branches. (**D**) Alterations in soma size (white dot, control; black dot, RF-EMF). Data are presented as mean ± SEM. Statistical significance is analyzed by using Student’s *t*-test (**B**–**D**). * *p* < 0.05 and ** *p* < 0.01 vs. control (control, *n* = 19–25 cells; RF-EMF, *n* = 19–28 cells). Scale bar, 50 μm.

**Figure 4 ijms-25-08589-f004:**
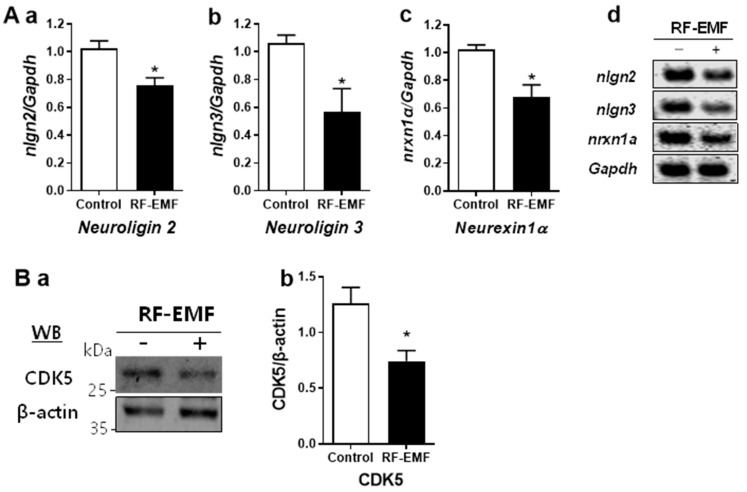
Synapse formation-related genes and proteins were altered in the cerebral cortexes of mice following exposure to RF-EMF. (**A**(**a**–**d**)) The expression levels of *neuroligin2*, *neuroligin3*, and *neurexin1α* are analyzed by qRT-PCR. (**B**(**a**,**b**)) The expression levels for cyclin-dependent kinase 5 protein in the cerebral cortex of mice after 4 weeks of exposure to RF-EMF. The protein level was normalized with β-actin. Data are expressed as mean ± SEM. Statistical significance is evaluated using Student’s *t*-test. * *p* < 0.05. vs. control (control, *n* = 5 mice; RF-EMF, *n* = 5 mice).

**Figure 5 ijms-25-08589-f005:**
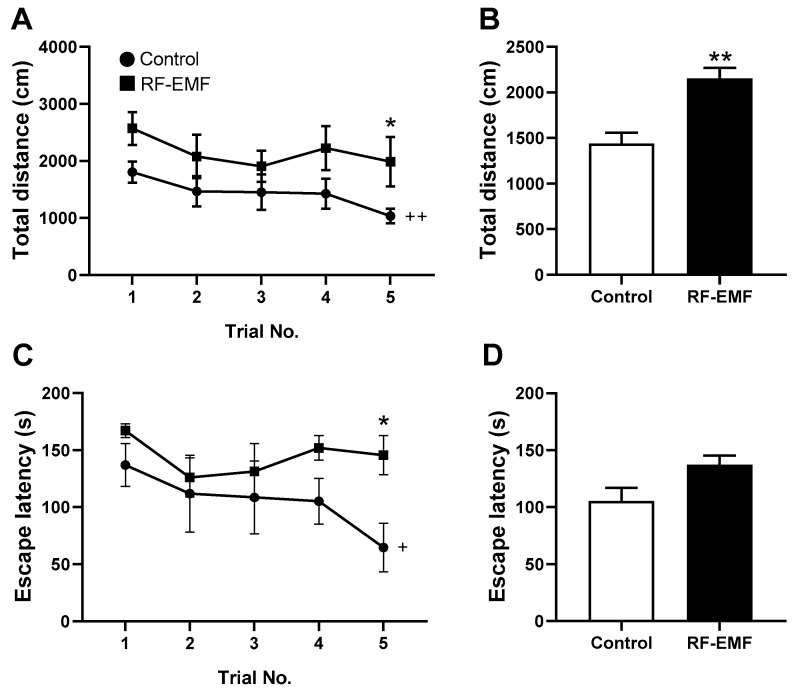
The evaluation of spatial learning and memory abilities in mice after RF-EMF exposure. Early postnatal mice were exposed to 1850 MHz RF-EMF at a specific absorption rate (SAR) of 4.0 W/kg for 5 h per day from postnatal day 1 [P1] to P28, followed by assessment using the Morris water maze test. (**A**) The total distance from trial 1 to trial 5 in the control and RF-EMF. (**B**) Average of total distance (trials 1–5). (**C**) Escape latency from trial 1 to trial 5 in control and RF-EMF. (**D**) Average of total escape latency (trials 1–5). Data are expressed as mean ± SEM. Statistical significance is evaluated using Student’s *t*-test. * *p* < 0.05, ** *p* < 0.05, + *p* < 0.05, and ++ *p* < 0.01 vs. control (control, *n* = 10 mice; RF-EMF, *n* = 8 mice). s = second.

## Data Availability

Data are contained within the article.

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
