# Peer review of "Exposure to Radiofrequency Induces Synaptic Dysfunction in Cortical Neurons Causing Learning and Memory Alteration in Early Postnatal Mice"

_ijms, 2024, doi:10.3390/ijms25168589_

Round 1

Reviewer 1 Report

Comments and Suggestions for Authors

fascinating manuscript; well designed, results are well presented

some minor points:

1- avoid the use of abbreviations in keywords, titles, fig legend

2-  In discussion; please link your results with previous studies especially that was stated in the introduction there is contradictory evidence on the g the biological effects of RF-EMF exposure on the central nervous system 

3- Please add the dilution of antibodies used in western blot and immunohistochemistry

4- add the total number of animals used

5- Any mortality?! 

Comments on the Quality of English Language

minor errors 

Author Response

Response to reviewer comments:

We really appreciate the reviewers for the careful review of our manuscript. We have revised the manuscript by addressing all the reviewers’ comments in a point-by-point response and indicated all the amends on the revised manuscript.

Reviewer 1

fascinating manuscript; well designed, results are well presented

  • Thank you for your very positive comment.

some minor points:

1- avoid the use of abbreviations in keywords, titles, fig legend

  • Thank you for your careful comment. We now avoid all abbreviations in those section in the revised manuscript.

2- In discussion; please link your results with previous studies especially that was stated in the introduction there is contradictory evidence on the g the biological effects of RF-EMF exposure on the central nervous system

  • Thank you for your careful comment. We now added the statement about the contradictory evidence on the biological effects of RF-EMF exposure on the central nervous system in discussion in the revised manuscript.

3- Please add the dilution of antibodies used in western blot and immunohistochemistry

  • We now added the dilution of antibodies used in our revised manuscript.

4- add the total number of animals used

  • We now added the total number of animals used in this study in our revised manuscript.

5- Any mortality?!

  • Thanks for your question about the mortality by RF-EMF exposure in this study. There were no animals were found to die under the experimental conditions used in this study. Now, we further clarify this at the methods in our study.

Reviewer 2 Report

Comments and Suggestions for Authors

Reviewer comments are attached as a PDF document.

Comments on the Quality of English Language

Overall, the linguistic style and grammar of the manuscript is acceptable, there a few corrections needed (see specific comments in the uploaded file, among the other comments concerning the content).

Author Response

Response to reviewer comments:

We really appreciate the reviewers for the careful review of our manuscript. We have revised the manuscript by addressing all the reviewers’ comments in a point-by-point response and indicated all the amends on the revised manuscript.

Reviewer 2

The paper by Kim and colleagues presents some interesting new data on the effects of radiofrequency electromagnetic fields on the development of rodent CNS in the early postnatal period. They give evidence that 28-day exposure to RF field of mouse pups decreases the number of dendritic spines in the prefrontal cortex, along with the expression of several proteins important in synapse functioning. Treated mice also displayed altered learning capacity in the Morris water maze. In addition, cell cultures prepared from cortical neurons showed decreased development and synapse formation in response to in vitro RF-EMF treatment. These data increase our understanding about the risks of RF-EMF exposure in humans. Overall, the manuscript is well organized, the experiments are planned and performed correctly, but the description of the methods lacks some important details. Also, the background information and the discussion of the results would benefit from some deeper interpretation with more aspects. My comments are listed below. If the authors respond to these remarks and complete the manuscript accordingly, I recommend the publication in International Journal for Molecular Sciences.

  • We really appreciate the reviewer's very favorable comments on our manuscripts.

Introduction

  • The introduction is a bit short and simplified. Cortical dysfunction is only one factor and one brain part involved in disorders such as AD, ASD, ADHD. Synaptic connections and structure which were examined in the current study are also only one factor that may lead to altered brain function. Of course, it would be out of the scope of an experimental paper to present the whole complexity of the possible ways of brain function alterations, but it would be nice, if the authors would mention other brain areas, other processes, too.
  • Thanks for your kind suggestions on our manuscripts. Now, we mentioned briefly of other brain areas, other processes in our revised manuscript.

“Although, cortical dysfunction and alterations in synaptic connections are important aspects of brain disorders, a comprehensive understanding requires considering the interactions between neurotransmitter systems, neurodevelopmental processes, brain connectivity, inflammation, genetic and epigenetic factors, and environmental influences. Each of these components contributes to the complex etiology and manifestation of various brain related disease, including Alzheimer's disease, autism spectrum disorder, and attention deficit hyperactivity disorder, etc.”

  • p.1, line 39: The citation of the reference is somewhat incorrect. „Mobile phone use among children has become a major determinant of RF-EMF exposure [5].” – However, the cited study states that radiation originating from television and radio antennas (broadcast) and mobile phone base stations (downlink) has a much higher share in RF-EMF exposure than actual mobile phone use (uplink).
  • Thank you for your careful comment. We now added appropriate references in the revised manuscript.
  1. Mobile phone use and exposures in children, January 2005 Bioelectromagnetics Suppl 7(S7):S45-50
  2. Exposure to Radiofrequency Electromagnetic Field in the High-Frequency Band and Cognitive Function in Children and Adolescents: A Literature Review, Int J Environ Res Public Health. 2020 Dec; 17(24): 9179.

  • p.2, line 49-51. The part about the cerebral cortex could be better formulated. For example, instead of writing „is located in the outer brain”, I suggest writing „is located on the surface of the brain”. Also, the convoluted structure is characteristic only for certain mammalian species, including humans, but the experimental animal used here (mouse) is lissencephalic! So I would not emphasize so much about the sulci and gyri, rather highlight only the important functions of the neocortex in mammals. From the enumeration of the main functions, by the way, the motor functions are missing!
  • Reviewer’s points are absolutely correct. We have revised the manuscript in accordance with the reviewer’s suggestions.

“The cerebral cortex is a highly developed region situated on the surface of the brain [9]. The neocortex, a crucial part of the cerebral cortex in mammals, is involved in a range of essential functions including sensory perception, motor control, higher cognitive processes, and complex behaviors. It plays a pivotal role in integrating sensory information, facilitating executive functions such as decision-making and problem-solving, and supporting language and communication skills [10].”

  • p. 2, line 63-64: „The exposure regimen involved daily exposure to 1850 MHz RF-EMF at a SAR of 4.0 W/kg for 5 h per day over a period of 4 weeks.” – Why did the authors choose this exact frequency? How does the applied intensity relate to existing public health regulatory limits? These points should be included into this part.
  • Thank you for reviewer’s careful comments. To address the comments raised by reviewer, we now added a new paragraph in discussion section.

“In this study, the frequency of RF-EMF was determined by according to commonly used in actual telecommunication services. The 4th generation (4G) communication systems operate globally within the frequency range of 600 MHz to 2.5 GHz for Long-Term Evolution (LTE) services. In this study, a frequency of 1850 MHz was studied, as it corresponds to one of the most widely used bands for 4G LTE communication services in South Korea. In addition, the 4 W/kg SAR value we used in this study is the maximum permitted SAR exposure to normal users, standardized based on SAR-related international organizations and major countries (The SAR standards of SAR-related international organizations and major countries (National Radio Research Agency, https://rra.go.kr/en/sar/standard.do).”

Methods

  • According to literature sources, ICR mice suffer frequently from severe visual impairment, so they are not recommended for the Morris water maze task (https://www.sciencedirect.com/science/article/pii/S0166432802000207). Indeed, the swimming distances, escape latencies seem very long in the current study. Maybe another cognitive test, not relying so much on visual cues, would be better to assess the cognitive deficit evoked by RF-EM field. The authors should mention this limitation of the mouse strain in the manuscript.
  • Thank you for reviewer’s careful comments. We added this limitation of the mouse strain in methods section.

“According to the literature, ICR mice are known to experience significant visual impairments, which makes them unsuitable for tasks such as the Morris water maze (https://www.sciencedirect.com/science/article/pii/S0166432802000207). Therefore, there may be limitations in interpreting the water maze results in our study, and additionally, it may be recommended to employ an alternative cognitive test that does not heavily depend on visual cues to more accurately assess the cognitive deficits induced by RF-EM fields.”

  • P.9., line 251. “After 3 weeks of feeding, dams were separated from the pups and continued to be exposed to RF-EMF for a week.” – this sentence can be misunderstood, like the dams received continued treatment and not the pups.
  • We are very sorry for confusing the reviewer. The sentence was modified to convey the meaning clearly.

After 3 weeks of feeding, the dams were separated from their pups, and only the pups continued to be exposed to RF-EMF for an additional week.”

  • P. 9, line 255. „same number of pups were provided to each dam” – so was there a foster mother paradigm? Or do the authors simply mean they limited the litter size by eliminating some pups? This is not clear from the description.
  • We are very sorry for confusing the reviewer. The sentence was modified to convey the meaning clearly.

“The number of pups was matched each group by eliminating some pups to minimize the weight difference of the pup between experimental group.”

  • Chapters 4.3, 4.4 and 4.5 – exposure conditions of the cell cultures are specified 3 times, this is an unnecessary repetition. It would be sufficient to state only once that the cultures “were exposed to 1760 MHz RF-EMF for 5 h daily from days in vitro (DIV) 0 to DIV 9”.
  • We now removed the unnecessary repetition in the revised manuscript.

  • Chapters 4.7, 4.8, 4.9: When were the mice sacrificed after the 28-day treatment? Was there a delay after the end of treatment?
  • After exposure to RF-EMF for 28 days, mice were sacrificed on the next morning (postnatal day 29) and brain samples were collected.

  • Why was CDK-5 quantified from hippocampus, while neurexins, neuroligin from cortex? (This may only be an error, because in the Results section, all proteins are connected to the cortex.)
  • Thank you for the reviewer’s comment. It was an typo, we now correct in the revised manuscript.

“Sham-exposed or RF-EMF-exposed mice were rapidly sacrificed, and prefrontal cortex area was rapidly dissected.”

  • About the Morris water maze: the size of the platform seems unusually big (17*48 cm). Was this really the platform size?
  • We are very sorry for confusing the reviewer. Actually, the platform has a diameter of 17 cm and a height of 48 cm. The size of the platform was modified in the revised manuscript.

“A platform (diameter 17cm) submerged 1.5 cm under the water surface was placed on the center of one of the four imaginary quadrants of the tank and maintained in the same position during all trials.”

  • Chapter 4.7: Quantification of dendritic spines in cerebral cortex with electron microscopy: I would expect that spine density varies according to cortical areas and layers. Was the sampling site standardized in some way?
  • Thank you for reviewer’s careful comments. As the reviewer pointed out, it is true that each cortex area and layer has various densities. However, in order to minimize these differences between each experimental group, we tried to secure the same parts as much as possible when preparing electron microscope samples, secured various sections, and randomly acquired multiple electron microscope images. In the images obtained in this way, the dendritic spine was analyzed to ensure that the comparison between the RF-EMF exposure group and the control group.

We added this limitation in the method section in the revised manuscript.

  • Page 8, line 207. states using the prefrontal cortex for the study of synaptic protein expression with PCR, however, according to the Methods chapter, the whole cortex was isolated!
  • In this experiment, the whole brain was separated from the mouse and then cut again for each part of the brain on ice. Since the cortex area is a relatively large area in the brain, the prefrontal cortex area was additionally excised and used in this experiment.

  • The description of the method used for the counting of the dendritic spines is missing! This is an important point which should be amended.
  • Thank you for the reviewer’s comment. We added the methodology for counting of the dendritic spines in Materials and Methods.

To measure for number of dendritic spines, samples were immediately prepared with control mice (n = 4) and RF-EMF exposed mice (n = 4). We generated 6-7 images per mouse and counted the number of different types (thin, stubby and mushroom) of dendritic spines in 28 images of control group and 25 images of RF-EMF exposed group. The average value of each type of dendritic spines measured was shown.

Results

  • Fig. 1 B: it is unclear how the number of spines is quantified (number/field of vision or number / a specific area?). Should the total number of spines not be equal to the sum of the numbers of the 3 subtypes (thin +stubby +mushroom)? Please clarify these data.
  • To measure for number of dendritic spines, samples were immediately prepared with control mice (n = 4) and RF-EMF exposed mice (n = 4). We generated 6-7 images per mouse and counted the number of different types (thin, stubby and mushroom) of dendritic spines in 28 images of control group and 25 images of RF-EMF exposed group. The average value of each type of dendritic spines measured was shown.

  • P. 4, line 129-130. “The soma size of RF-EMF-exposed cultured cortical neurons generally increased on DIV 3, 7, and 9, but not on DIV 5 (Figure 3D).” – It would be better to write that it was higher than in controls. Because the soma size was compared to that of control cultures, and the development in time was not directly analyzed (the word „increase” suggests a change in time).
  • Thank you very much for your thoughtful advice. We modified it as follows: The soma size of RF-EMF-exposed cultured cortical neurons generally higher than in controls on DIV 3, 7, and 9, but not on DIV 5 (Figure 3D).

  • Fig. 3. The small diagrams in part B do not have a sufficient resolution
  • In the revised manuscript, the image was changed to a higher resolution.

  • Fig. 4 panel Ad: it would be nice to have the kDa landmarks here as well, like in panel Ba. In panel Ba, the CDK part in the manuscript is not visible, although the image is OK in the separately attached figure file.
  • 4 panel Ad is not a protein band from western blotting, but a product band of approximately 100 bp from the semi-quantitative RT-PCR, and the signal intensity is visualized using 1.5% agarose gel electrophoresis. It is clarified in 5.8 Quantitative RT-PCR in Materials and Methods.

  • Page 6, line 164. English editing should be performed here, instead of “chase program”,

the term “tracking program” should be used for Noldus Ethovision.

  • Thank you for reviewer’s careful comments. It was revised according to the reviewer's suggestion.

“After training, five actual trials were conducted. Swimming distance (cm) and escape latency (s) were measured using a chase program.”

  • Fig. 5. here the panels are labelled as A,B,C,D, but in the legend there is Aa to Bb. For panels C and D, the legend is: sum of distances/times for the 5 trials. But looking at the numbers, it seems rather the average than the sum….

  • Thank you for reviewer’s careful comments and we are very sorry for confusing the reviewer. Reviewer comments are correct. Fig. 5. B and D are an average of 1-5 trials. It was revised according to the reviewer's comments.

Discussion

The discussion seems a bit weak, with very general statements. Also, there are some important points which should be discussed, as follows.

  • The same field intensity was applied to whole mice and cell cultures. However, in case of the neuronal cultures, a cell monolayer is directly exposed, without the shielding effect of the skull etc. Is the exposure of the neurons similar in the two conditions? The test systems themselves differ much in complexity, which may affect the vulnerability of the neurons.
  • This is a very good comments regarding our research.

The RF-EMF exposure devices used in this experiment are separated equipment. In our laboratory, we have installed and are operating a RF-EMF exposure device for mice and for cell, respectively.

As the reviewer pointed out, it is true that RF-EMF exposed directly in cells and through the skull in animals. However, since SAR (specific absorption rate), which is used as the exposure amount related to RF-EMF exposure, is set based on the absorbed energy value. It is determined by the absorbed value, not the applied value by each RF-EMF device. Therefore, it is believed that the actual amount of electromagnetic wave delivered will be the same in cell and brain of animal.

  • In case of the in vitro treatment, the thermal effect was compensated with water cooling. What about the case of the in vivo treatment? Was cooling not necessary here?
  • That's a very good point and we really appreciated to the reviewer.

In the study of electromagnetic effects, thermal effect is a very important issue. As the reviewer pointed out, the cell exposure device has an additional cooling system to keep the temperature from rising by RF-EMF exposure. However, in animal experiments, there is no additional cooling system but the experiments are conducted in a laboratory where constant temperature and humidity are maintained. Especially, in animals, it is thought that some heat by RF-EMF exposure will be cooling down, due to the homeostasis effect of the circulatory system of animal. Additionally, when observing the change in body temperature of mice immediately after exposure to RF-EMF under these experimental conditions, no increase in body temperature is observed. Therefore, we believe that this experiment was conducted with minimal thermal effects in both animal and cell experiments.

  • The data after in vivo treatment show the results of a prolonged RF field exposure during early postnatal development, while in the in vitro study, the authors could follow the alterations during the treatment.
  • Thank you for reviewer’s careful comment. As the reviewer pointed out, sampling was conducted on the 28th day after RF-EMF exposure was completed in the mouse experiment, but cells were observed continuously for 9 days of RF-EMF exposure.

We think that the two experiments are complementary in their study of the effects of RF-EMF on the central nervous system. The effects on nerve cells in mice exposed to RF-EMF were measured in terms of final results and behavioral changes. These results show that RF-EMF exposed during early neurodevelopmental stage can affect the cerebral cortex of mouse. However, in experiments using cultured cortical neurons, it helps to figure out the possible mechanisms about the possible effects on neurons (changes in synapse density and growth, inhibition of neurite outgrowth, etc.).

Another excuse, a dam after giving birth is very sensitive and aggressive, so we tried to keep contact either dam or pups to a minimum, to prevent infanticide in rodents. In addition, at the early life of mouse, the brains are so small that it is difficult to distinguish which parts of the brain are needed.

  • RF-EM field effects are probably not brain area specific, so maybe the parts about PFC and in which disorders it is implicated are not so relevant. Similar effects may be expected in sensory and motor areas.
  • We completely agree with the reviewer’s points. In addition to the observed effects on the prefrontal cortex (PFC) that were the primary focus of this study, it is absolutely possible that sensory and motor areas may also be influenced either structurally or functionally. However, this study was specifically concentrated on the possible effects on PFC. Future research will address the potential impact on additional body parts as suggested by the reviewer.

  • Putative mechanisms of effect of the RF-EM field on CNS development and functions should be briefly discussed, with appropriate references, either in the Discussion or the Introduction part! Also, the relevance of the applied field intensity should be stated (is it realistic or very strong compared to everyday conditions?).
  • Thank you for reviewer’s careful comments. We added the possible mechanisms of effect of the RF-EMF on CNS development and functions in discussion section.

“Although several studies are in progress, and a clear mechanism has not yet been revealed yet, several putative mechanisms have been proposed. RF-EMF can cause localized heating of tissues, potentially affecting neural tissue if exposure is high enough (Okechukwu, 2020; Christopher et al., 2021). Also, some studies suggest that RF-EMF might influence CNS function through oxidative stress (Meral et al., 2007; Consales et al., 2012), neurotransmitter release alteration (Kim et al., 2019; Hu et al., 2021), gene expression (Zhao et al., 2007), inducing weak electrical currents in neural tissues (Lai, 2012), affecting neuronal cell proliferation or apoptosis (Liu et al., 2012), and impacting blood-brain barrier permeability (Stam, 2010). While there are numerous hypotheses about how RF-EM fields might influence CNS development and function (Kaplan et al., 2016), definitive conclusions are challenging due to variability in study designs, exposure levels, and biological responses. More research is needed to clarify these mechanisms and their implications for health.”

Regarding intensity, as we answered above, the 4 W/kg SAR value we used in this study is the maximum permitted SAR exposure to normal users, standardized based on SAR-related international organizations and major countries. Therefore, the exposure amount used in this study is not an impossible value in reality, but it is expected to occur very rarely because it corresponds to the highest limit.

  • Reference [16] seems totally out of scope of the manuscript, it should be omitted.
  • Thank you for your careful comment. We now removed the reference [16] in the revised manuscript
